# Short-term dietary fiber interventions produce consistent gut microbiome responses across studies

Cynthia I. Rodriguez,[1] Kazuo Isobe,[2] Jennifer B. H. Martiny[1]

**ABSTRACT** The composition of the human gut microbiome varies tremendously among individuals, making the effects of dietary or treatment interventions difficult to detect and characterize. The consumption of fiber is important for gut health, yet the specific effects of increased fiber intake on the gut microbiome vary across studies. The variation in study outcomes might be due to inter-individual (or inter-population) variation or to the details of the interventions including the types of fiber, length of study, size of cohort, and molecular approaches. Thus, to identify generally (on average) consistent fiber-induced responses in the gut microbiome of healthy individuals, we re-analyzed 16S rRNA sequencing data from 21 dietary fiber interventions from 12 human studies, which included 2,564 fecal samples from 538 subjects across all interventions. Short-term increases in dietary fiber consumption resulted in highly consistent gut bacterial community responses across studies. Increased fiber consumption explained an average of 1.5% of compositional variation (vs 82% of variation attributed to the individual), reduced alpha-diversity, and resulted in phylogenetically conserved responses in relative abundances among bacterial taxa. Additionally, we identified bacterial clades, at approximately the genus level, that were highly consistent in their response (on average, increasing or decreasing in their relative abundance) to dietary fiber interventions across the studies.

**IMPORTANCE** Our study is an example of the power of synthesizing and reanalyzing 16S rRNA microbiome data from many intervention studies. Despite high inter-individual variation of the composition of the human gut microbiome, dietary fiber interventions cause a consistent response both in the degree of change and the particular taxa that respond to increased fiber.

**KEYWORDS** gut microbiome, fiber, meta-analysis

Dietary fibers are carbohydrates that resist digestion by the small intestine and have positive health impacts on humans (1). High-fiber diets are associated with health benefits such as increased nutrient absorption, production of beneficial metabolites, improved immune responses, and amelioration of various diseases including obesity, diabetes, allergies, and others (2–6). To understand the influence of dietary fiber on the gut microbiota, researchers have performed dietary interventions using a variety of fiber compounds on humans (7–9).

Experiments that increase fiber intake in humans often result in shifts in bacterial composition in the gut microbiome. Hereafter, we use the term microbiome to refer to the bacterial communities, while understanding that this is a simplification as it does not include viruses, archaea, and microbial eukaryotes. For example, fiber interventions including inulin and gala-oligosaccharides often report an increase of *Bifidobacterium* and *Lactobacillus* taxa in the gut, genera known as lactic acid producers and carbohydrate degraders (10–12). In addition, plant-based diets (with high fiber content) enrich

Address correspondence to Cynthia I. Rodriguez, cirodri1@uci.edu.

The authors declare no conflict of interest.

See the funding table on p. 14.

for the genera *Ruminococcus* and *Prevotella*, which degrade and ferment complex dietary carbohydrates (13–18). However, while some bacterial responses seem to be consistent across fiber interventions, other studies report contradictory trends (8, 19, 20). For instance Tian et al. (19), found no increases in the taxa mentioned above, but instead observed a decrease in *Ruminococcus*, and Whisner and colleagues (2018) found that *Ruminococcus* were more abundant in a group of college students that consumed low fiber foods (20).

Such contradictions are not necessarily surprising, as comparing results across any type of microbiome intervention comes with at least three challenges. The first obstacle is heterogeneity in study design and technical approaches. For fiber interventions in particular, studies vary in the types of fiber compounds used, intervention lengths, and population sizes. Moreover, differences in molecular approaches and in downstream bioinformatic pipelines add technical variation to the characterization of microbiome composition that potentially obscures biological patterns across studies.

A second challenge is the high inter-individual variation of gut microbiome composition. This variation can be due to many factors such as host genetics, diet, medical conditions, pet ownership, and stool consistency just to mention some (21, 22). Such differences make comparing microbiome responses across individuals difficult, let alone across studies. Not only does the starting pre-intervention composition of the gut microbiome vary widely between individuals, but many operational taxonomic units (OTUs) are not shared among individuals within a study. As a result, the variation in gut composition explained by an intervention will be typically small relative to inter-individual variation and, thus, may be difficult to detect and characterize.

Finally, comparing taxa across studies can be difficult. Not only can OTUs of bacterial sequences be defined differently across studies (e.g., at different cutoffs such as 100%, 99%, and 97% sequence similarity), but the results are often summarized at different taxonomic levels. For instance, some studies may report changes in relative abundance in terms of phyla (e.g., Actinobacteria), whereas others by family or genus (e.g., *Bifidobacteriaceae*/*Bifidobacterium*). Moreover, the consistency of responses of finer-scale taxa within a reported taxonomic level is often unclear as an individual's gut typically contains several strains and/or species within the same genus (23). For instance, healthy adults can harbor up to six species of *Bifidobacterium* at any one time (24, 25). However, most studies report only the most responsive OTUs and/or changes in relative abundances lumped at a broader taxonomic level. Thus, variation in the responses of finer-scale taxa (and in their distribution among individuals) might contribute to inconsistent results among intervention studies.

Although some of the above-mentioned discrepancies cannot be modified for past studies (e.g., study design and sequencing processes), there are avenues to improve comparisons of past results across interventions. One approach is to reanalyze the data in a consistent manner and then use phylogenetic information to organize the biological variation. Specifically, the raw data (e.g., 16S rRNA sequencing reads) can be uniformly processed using similar bioinformatic pipelines, threshold parameters, and statistical analyses. Then, phylogenetic placement of the sequences can be used to precisely compare compositional shifts across studies. Furthermore, this approach can shed light on the phylogenetic depth of the response to the intervention (26, 27). If lineages within a clade respond in a similar (positive or negative) manner to an intervention, then this phylogenetic signal provides a hypothesis about how microbiome composition may respond in other human populations, even if the fine-scale taxonomic composition (i.e., the precise OTUs) among populations is highly divergent.

Here, we took this analytical approach to investigate the consistency of fiber-induced changes in the gut microbiome of healthy individuals by re-analyzing 16S rRNA sequencing data from 21 dietary fiber interventions. Ideally, the interventions would include a range of fiber types; however, the studies reflected a bias toward readily available supplements. Most of the interventions involved dietary supplements of alpha-glycans (e.g., resistant starch, polydextrose) and fructans (fructooligosaccharides

and inulin). Only one intervention involved whole foods, whereas others used minimally digestible supplements.

We hypothesized that short-term increases in fiber intake would result in consistent changes in microbiome composition even though many aspects of the fiber interventions varied, including the type and amount of fiber and the duration of the intervention. To test this hypothesis, we assessed three features of each intervention: (i) changes in bacterial alpha-diversity after the fiber intervention, (ii) the amount of compositional variation (beta-diversity) explained by the fiber intervention relative to that of between individuals, and (iii) taxa responses in a phylogenetic context to identify consistent fiber-responding clades.

## MATERIALS AND METHODS

### Study inclusion criteria

The search for studies has been previously described in the data description paper by Rodriguez et al. (28). Briefly, we performed a keyword search of published literature on 9 May 2020 through the PubMed search engine (keywords: dietary, fiber, and microbiome) under the Best Match algorithm recommended by PubMed. The search yielded 977 abstract hits from 2010 to 2020 (https://pubmed.ncbi.nlm.nih.gov/). We also searched through all the records available in the database of open-source microbial management site Qiita (29) on 7 April 2020 and found 528 microbiome studies including human and animal studies (https://qiita.ucsd.edu). From both sources, each abstract was carefully read to select studies with fiber interventions in healthy humans that included 16S rRNA amplicon sequencing data from fecal microbial communities ($n = 34$). We excluded studies in animals and unhealthy humans. Corresponding authors and first authors were contacted up to four times requesting their sequencing data and metadata when not publicly available. We were able to obtain 16S rRNA amplicon sequencing reads and their corresponding metadata from 12 studies and within these, 5 conducted diet interventions with different types of dietary fibers and/or food sources (Tables 1 and 2). When this was the case, the sequence data in each study were divided by the fiber intervention, resulting in a total of 21 intervention experiments (Table 3). For example, if one study conducted separate interventions with inulin and psyllium, the data set was divided into two. We named each of the interventions as: Last name of the first author in the publication, followed by the year the study was published, continuing with the region of the 16S rRNA bacterial gene that was amplified, with the addition of the fiber used in the study (e.g., Baxter_2019_V4_potato). In total, we analyzed 16S rRNA sequencing data from 2,564 fecal samples derived from 538 subjects across all study interventions (Table 3). All studies included in this analysis adhered to their relevant ethical standards and received consent from their human participants to collect and share the data. For more information regarding guidelines for study procedure and trial registration numbers, we refer our readers to the individual studies referenced in Table 1. Supplemental information (e.g., manufacturer) regarding the specific fiber composition of each dietary intervention can be found in the data description paper by Rodriguez et al. (28) in their Table 4.

### Sequencing processing

To compare the sequences directly across studies, we obtained the raw sequencing reads for each study and processed them in a similar manner. First, we assessed the quality of the 16S rRNA sequencing data using FastQC software version 0.11.8 (30). The sequencing reads were cleaned from poor quality sequences using the Fastp program version 0.20.0 (31). The cleaned sequences were imported into the QIIME2 platform version 2020.11.1 (32), and primers were removed using Cutadapt plugin (33) when necessary. We then denoised the reads using DADA2 plugin (34), obtaining exact sequence variants (ESVs) tables depicting the number of reads per sample for each ESV. Besides using DADA2 to denoise data, no further filtering was done to remove rare ESVs.

**TABLE 1** Data repositories for individual fiber intervention studies

| Study name | Repository for raw data | Accession number for raw data | Sequencing platform used | Single- or paired-end data |
|---|---|---|---|---|
| Baxter_2019_V4 | NCBI Sequence Read Archive | SRP128128 | Illumina MiSeq | Paired |
| Dahl_2016_V1V2 | NCBI Sequence Read Archive | SRP403421 | Illumina MiSeq | Paired |
| Deehan_2020_V5V6 | NCBI Sequence Read Archive | SRP219296 | Illumina MiSeq | Paired |
| Healey_2018_V3V4 | NCBI Sequence Read Archive | SRP120250 | Illumina MiSeq | Paired |
| Hooda_2012_V4V6 | NCBI Sequence Read Archive | SRP403421 | 454/Roche pyrosequencing | Single |
| Kovatcheva_2015_V1V2 | NCBI Sequence Read Archive | SRP062889 | 454/Roche pyrosequencing | Single |
| Liu_2017_V4 | European Nucleotide Archive | PRJEB15149 | Ion Torrent | Single |
| Morales_2016_V3V4 | NCBI Sequence Read Archive | SRP403421 | Illumina MiSeq | Paired |
| Rasmussen_2017_V1V3 | NCBI Sequence Read Archive | SRP106361 | 454/Roche pyrosequencing | Single |
| Tap_2015_V3V4 | European Nucleotide Archive | PRJEB2165 | 454/Roche pyrosequencing | Single |
| Vandeputte_2017_V4 | European Genotyping Agency | EGAS00001002173 | Illumina MiSeq | Paired |
| Venkataraman_2016_V4 | NCBI Sequence Read Archive | SRP067761 | Illumina MiSeq | Paired |

Next, the taxonomic classification of the reads was also performed in the QIIME2 platform by training the SILVA version 132_99_16S (35) and the Genome Taxonomy Database (GTDB) version bac120_ssu_reps_r95 (36) databases to each respective study based on the primers that were originally used. The SILVA database was used to remove chloroplast and mitochondrial DNA. Then, the cleaned reads were assigned to a final bacterial taxonomic group using the GTDB trained database. Only reads classified to the phylum level and beyond were kept in the ESV tables. All processed data sets described have been deposited to Figshare (https://doi.org/10.6084/m9.figshare.21295352), except for Vandeputte and colleagues (37), whose raw 16S rRNA data can be accessed through the European Genotyping Agency (EGAS00001002173).

## Bacterial community composition responses to individual fiber interventions

For the analysis of individual fiber interventions, we used the forward reads (for uniformity) from all the studies and imported the data into R (version 4.0.2) for rarefaction to normalize for sequencing depth before the alpha- and beta-diversity analyses. We calculated rarefied ESV tables through randomized sampling sequences without replacement for 1,000 iterations, using the highest sequencing depth possible for each data set (Table 3). Although there is some controversy about the best method to standardize for sequencing depth, recent work comparing standardization techniques concluded that rarefaction provides a robust method for microbiome data (38). For each study, we only used samples from the fiber intervention treatments and excluded samples from other treatments (e.g., drugs, white wheat bread, maltodextrin-controls).

We tested for differences in alpha-diversity using Shannon and Simpson indices, and overall bacterial richness (calculated as the total number of bacterial taxa per sample) before and after fiber interventions using the rarefied ESV tables via vegan package, version 2.6-2, and paired-$t$ tests in R, version 4.0.2. When multiple timepoints were collected from the same subject before and after the fiber intervention, we used only two timepoints (the earliest timepoint before, and the latest sample after the intervention) to allow for paired-$t$-test analyses.

To test differences in bacterial community composition (beta-diversity), we ran permutational multivariate analysis of variance (PERMANOVA) on Bray-Curtis dissimilarity matrices including all timepoints available for each study, grouping the fiber intervention timepoints as "before" vs "after" when multiple samples per individual were available. We decided to use Bray-Curtis dissimilarity metric as we were interested in accounting for the relative abundances of taxa across samples, rather than giving them equal weight (e.g., a presence/absence metric such as Jaccard). These differences are important when looking into microbiome changes from the same individual across time as the same taxa can be present before the dietary intervention but at different abundances. To construct these matrices, we averaged dissimilarity matrices created from rarefied and square-root

**TABLE 2** Summary of data sets collected including fiber type, grams of fiber used, duration of the intervention, number of timepoints for fecal collections, number of subjects, and total number of fecal samples per study[a]

| Study name | Interventions (#) | Fibers used | Amount of fiber (grams) | Duration (days) | Timepoints (#) | Subjects (#) | Samples (#) |
|---|---|---|---|---|---|---|---|
| Baxter_2019_V4 | 3 | Resistant starch from potatoes (RPS), resistant starch from maize (RMS), and inulin from chicory root | 20–40 | 14 | 8 | 175 | 1,205 |
| Dahl_2016_V1V2 | 3 | RS-4-A, RS-4-B, RS-4-C—resistant potato starches (RS type 4) | 30 | 14 | 4 | 53 | 212 |
| Deehan_2020_V5V6 | 3 | Cross-linked tapioca, cross-linked potato, and maize—resistant starches (RS type 4) | Increasing from 0 to 10, 20, 28 35, and 50 | | 5 | 40 | 200 |
| Healey_2018_V3V4 | 1 | 50:50 inulin to fructo-oligosaccharide (FOS) | 16 | 21 | 4 | 34 | 134 |
| Hooda_2012_V4V6 | 2 | Polydextrose and soluble corn fiber | 21 | 21 | 3 | 10 | 28 |
| Kovatcheva_2015_V1V2 | 1 | Kernel-based bread (KBB) and white-wheat bread (WWB) | 37.6 and 9.1 | 3 | 3 | 20 | 60 |
| Liu_2017_V4 | 2 | Fructooligosaccharides (FOS) and galactooligosaccharides (GOS) | 16 | 14 | 4 | 35 | 132 |
| Morales_2016_V3V4 | 1 | Oligofructose | 16 | 7 | 2 | 41 | 82 |
| Rasmussen_2017_V1V3 | 2 | Starch-entrapped microspheres and psyllium | 9 and 12 | 84 | 2 | 41 | 82 |
| Tap_2015_V3V4 | 1 | Dietary fiber meals | 10 and 40 | 5 | 4 | 19 | 76 |
| Vandeputte_2017_V4 | 1 | Inulin | 12 | 28 | 4 | 50 | 196 |
| Venkataraman_2016_V4 | 1 | Resistant starch (unmodified potato starch; RS type 2) | 48 | 17 | 8 | 20 | 157 |

[a]The interventions column refers to the dietary fiber interventions that we included in our analysis per study. For each study, we only used samples from the fiber intervention treatments and excluded samples from the controls and other low fiber treatments (e.g., white-wheat bread).

**TABLE 3** Summary of the samples included and of the alpha- and beta-diversity results by fiber intervention[a]

| Study | Samples (#) | Subjects (#) | Rarefaction depth (# reads) | Alpha-diversity | | | Beta-diversity | |
|---|---|---|---|---|---|---|---|---|
| | | | | Shannon | Simpson | Richness | Subject variation explained (%) | Fiber variation explained (%) |
| Baxter_2019_V4_himaize (RMS) | 313 | 43 | 4,891 | ↓ **significant** | ↓ **significant** | ↓ **significant** | 86 | **0.2** |
| Baxter_2019_V4_inulin | 365 | 50 | 4,546 | ↓ n.s. | ↓ n.s. | ↓ n.s. | 84 | **0.6** |
| Baxter_2019_V4_potato (RPS) | 273 | 43 | 4,622 | ↑ n.s. | ↑ n.s. | ↓ n.s. | 86 | **0.7** |
| Dahl_2016_V1V2_potato-RS4A (RPS) | 34 | 17 | 16,289 | ↓ n.s. | ↓ n.s. | ↓ n.s. | 89 | **1.0** |
| Dahl_2016_V1V2_potato-RS4B (RPS) | 36 | 18 | 15,957 | ↓ n.s. | ↓ n.s. | ↓ **significant** | 88 | **1.0** |
| Dahl_2016_V1V2_potato-RS4C (RPS) | 36 | 18 | 8,135 | ↓ n.s. | ↓ n.s. | ↓ **significant** | 87 | 0.8 |
| Deehan_2020_V5V6_maize-RS4 | 50 | 10 | 27,488 | ↓ **significant** | ↓ **significant** | ↓ **significant** | 87 | **1.0** |
| Deehan_2020_V5V6_potato-RS4 | 50 | 10 | 18,744 | ↓ n.s | ↓ n.s | ↓ n.s | 85 | 0.5 |
| Deehan_2020_V5V6_tapioca-RS4 | 50 | 10 | 7,157 | ↓ **significant** | ↓ **significant** | ↓ **significant** | 79 | **1.0** |
| Healey_2018_V3V4_inulin-FOS | 68 | 34 | 6,014 | ↓ **significant** | ↓ **significant** | ↓ **significant** | 85 | 1.6 |
| Hooda_2012_V4V6_corn | 19 | 10 | 3,689 | ↓ n.s. | ↓ n.s. | ↓ n.s. | 75 | 4.4 |
| Hooda_2012_V4V6_polydextrose | 19 | 10 | 2,966 | ↓ n.s. | ↓ n.s. | ↓ n.s. | 76 | **4.6** |
| Kovatcheva_2015_V1V2_kbb | 40 | 20 | 3,642 | ↑ n.s. | ↑ n.s. | ↓ n.s. | 86 | 0.6 |
| Liu_2017_V4_FOS | 66 | 34 | 1929 | ↓ n.s. | ↓ n.s. | ↓ **significant** | 80 | 0.9 |
| Liu_2017_V4_GOS | 66 | 34 | 1465 | ↓ n.s | ↓ n.s | ↓ **significant** | 78 | **1.5** |
| Morales_2016_V3V4_oligofructose | 22 | 11 | 66061 | ↓ n.s | ↓ n.s | ↓ **significant** | 88 | **1.6** |
| Rasmussen_2017_V1V3_SM12 | 30 | 15 | 3217 | ↓ n.s | ↓ n.s | ↑ n.s | 74 | 1.6 |
| Rasmussen_2017_V1V3_psyllium | 24 | 12 | 1197 | ↓ n.s | ↓ n.s | ↓ n.s | 76 | 2.6 |
| Tap_2015_V3V4_dietary-fiber-meals | 38 | 19 | 1021 | ↓ n.s | ↓ n.s | ↓ n.s | 71 | 1.4 |
| Vandeputte_2017_V4_inulin | 96 | 49 | 7912 | ↓ **significant** | ↓ **significant** | ↓ n.s | 83 | **0.7** |
| Venkataraman_2016_V4_potato (RPS) | 157 | 20 | 2167 | ↓ n.s | ↑ n.s | ↓ **significant** | 84 | **0.9** |
| Average | | | | | | | 82 | 1.4 |
| **Average—significant only** | | | | | | | 82 | 1.5 |

[a]We note the number of samples and subjects per intervention and the rarefaction depth used for the normalization of each data set for alpha- and beta-diversity analysis. The alpha-diversity column represents the results of the comparison between two timepoints (before vs after fiber intervention) for Shannon, Simpson, and richness indices; the arrow direction represents an increase (upward) or decrease (downward) in alpha diversity after the intervention regardless of significance. The beta-diversity columns show the variation explained by either subject or the fiber interventions. n.s, not significant; bold indicates $P < 0.05$. RMS, resistant corn starch; RPS, resistant potato starch; RS4, type-IV resistant starches; KBB, kernel-based bread; FOS, fructooligosaccharides; GOS, galactooligosaccharides; SM, starch-entrapped microspheres (SM). We note that the results presented here may vary slightly from those reported in the original work due to differences in sample inclusion and methodology; for more information, refer to Materials and Methods.

**TABLE 4** ConsenTRAIT results for individual studies[a]

| Study | No. of OTUs | Significantly responding OTUs | No. OTUs (>1.5 fold change) | Positive responding OTUs (#) | Negative responding OTUs (#) | Positive $\tau_D$ | Negative $\tau_D$ |
|---|---|---|---|---|---|---|---|
| Baxter_2019_V4 (RMS, RPS, inulin) | 137 | 40 | 4 | 52 | 85 | **0.028** | 0.020 |
| Healey_2018_V4 (inulin-FOS) | 312 | 24 | 23 | 140 | 172 | **0.017** | **0.019** |
| Hooda_2012_V4 (corn, polydextrose) | 208 | 23 | 23 | 115 | 93 | 0.018 | **0.020** |
| Liu_2017_V4 (FOS & GOS) | 86 | 12 | 12 | 31 | 55 | 0.019 | 0.029 |
| Morales_2016_V4 (oligofructose) | 1,044 | 11 | 11 | 494 | 550 | **0.014** | **0.015** |
| Tap_2015_V4 (high fiber meals) | 128 | 4 | 4 | 82 | 46 | **0.026** | 0.015 |
| Vandeputte_2017_V4 (inulin) | 463 | 22 | 22 | 136 | 327 | **0.016** | **0.021** |
| Venkataraman_2016_V4 (RSP) | 179 | 23 | 11 | 88 | 91 | **0.023** | 0.021 |
| Average | | | | | | 0.020 | 0.020 |
| **Average—significant only** | | | | | | **0.021** | **0.019** |

[a]Number of OTUs is the number of taxa at 97% identity that were present after filtering, followed by the number of significantly responding OTUs and OTUs that significantly shifted at >|1.5|-fold change. The positive and negative responding taxa columns correspond to the OTUs used to build the phylogenetic trees, which were found through DESeq2 with a log2-fold change higher than zero or below zero, respectively. Bold numbers represent that $\tau_D$ values are significantly >0 ($P < 0.05$). RMS, resistant corn starch; RPS, resistant potato starch; FOS, fructooligosaccharides; GOS, galactooligosaccharides.

transformed (to minimize the influence of the most abundant taxa) ESV tables (1,000 iterations) (39). The PERMANOVA formula used in the R vegan package was as follows: adonis2.(bray.dist.matrix ~subject_id + fiber_timepoint, data = metadata, method= "bray", by= "term", permutations = 999); where fiber_timepoint specifies whether the sample was collected pre- or post-intervention. Thus, a significant main effect of fiber_timepoint indicates that the fiber intervention altered microbiome composition in a consistent way, and a significant main effect of subject_id, that individuals have a distinct microbiome composition. Note that although we expect microbiome composition to vary over time within an individual for reasons unrelated to the fiber intervention (40), such temporal variability would not produce a significant result for fiber_timepoint as the changes are not likely to be consistent across individuals (41). Moreover, to test for differences in compositional variation between the before and after fiber intervention groups we conducted a betadisper analysis using vegan in R, a PERMDISP-like test for the analysis of multivariate homogeneity. Only the Deehan_2020_V5V6_maize intervention showed a significant beta dispersion between the aforementioned groups (Table S1). Because 16S rRNA analyses can be sensitive to processing parameters and we sometimes used a subset of the samples to make similar comparisons across studies, the results presented here may vary slightly from those reported in the original work.

## Phylogenetic responses to dietary fiber

To conduct an in-depth phylogenetic analysis, we next considered only studies (8/12) that shared the V4 region of the 16S rRNA gene (Table 4) and re-processed their sequences to compare specific OTUs between studies including all their fiber interventions (26, 42). When available, we merged the forward and reverse V4 reads using BBmerge from BBMap Tools version 38.95 (43). Then, we extracted the same V4 region across the eight studies with Cutadapt version 3.5 using the V4 primer sequences (forward: GTGYCAGCMGCCGCGGTAA; reverse: GGACTACNVGGGTWTCTAAT) from the Earth Microbiome Project (44). To ensure that the sequences were properly extracted (e.g., read size = 250 bp), we visualized them using Geneious prime (version 2020.2.4; https://www.geneious.com/), FastQC version 0.11.9 and summarized the results with Multiqc, version 1.11. Then, the extracted reads (250 bp) were imported into QIIME2 (version 2020.11) as a single artifact. The q2-vsearch plugin in QIIME2 was used to dereplicate the sequences and cluster them at 97% identity. Because our goal was to make in-depth phylogenetic comparisons across studies, we used 97% dereplication identity rather than ESVs to simplify the complexity of the gut bacterial responses across

studies using different collection and sequencing methods. Based on previous research (26), a finer-scale assignment of OTUs (e.g., ESVs) results in too few overlaps in OTUs among the studies making it difficult to make comparisons across interventions. Finally, we filtered the OTU table by removing OTUs with low abundance (<10 summed across all samples) and/or those in less than 3 samples based on the assumption that these may not represent real biological sequences but rather are sequencing errors or PCR chimeras. We assigned taxonomy as described above for each individual study using the V4 primer sequences from the Earth Microbiome Project. The merged data were then divided into OTU tables for each study. Finally, to focus on the responses of common taxa distributed widely among individuals, we excluded OTUs that were present in less than 50% of the samples per study. This stringent cutoff ensured that the responses were not driven solely by sporadic differences in abundances from just a few individuals (26, 27).

To perform a standard differential abundance analysis of the OTUs, we first used Phyloseq version 1.34.0 (45) to convert the data to the standard phyloseq-class data object to be used in DESeq2. For each study, we used the non-rarefied data in DESeq2 to (i) normalize the data and (ii) calculate the $\log_2$-fold ratio of the normalized OTU abundances to identify OTUs significantly affected by fiber treatment ($P$ adjusted < 0.05) with $\log_2$-fold change cutoff 0 and > $|0.58|$ (1.5-fold change). We then averaged the $\log_2$-fold change responses across studies. OTUs with a $\log_2$-fold change higher than zero were considered to be positive responding taxa, whereas the OTUs with a negative $\log_2$-fold change were considered negatively responding taxa.

To assess the phylogenetic conservation of fiber responses, we selected only widespread OTUs (present in >3 studies where, as above, present means found in >50% of a study's samples) to ensure that the response trends were not driven by just one or two studies or just a handful of individuals in a study. We aligned these sequences in R using the Biostrings version 2.58.0 and DECIPHER version 2.18 (46) packages to create a neighbor-joining (NJ) tree with phangorn package version 2.5.3 (47) using the 16S ribosomal gene from *Methanosarcina barkeri* as an outgroup. The positive and negative responding taxa were assigned a 1 and a 0, respectively. We then ran ConsenTRAIT, with percent shared trait cutoff of 0.9, using the castor package version 1.3.5 (48) to identify consensus clades, clades that respond to fiber intervention in the same direction across studies and to calculate the average depth ($\tau_D$) of the conserved clades from the NJ phylogenetic tree we created. We used an NJ tree for the consenTRAIT analysis because the genetic scale of NJ trees roughly represents sequence dissimilarity and to compare trait depth to similar analyses (26, 27). Previous studies have also found that ConsenTRAIT results are robust regardless of phylogenetic reconstruction method (26, 27). To corroborate this, we built a maximum likelihood (ML) tree with 100 bootstrap replications with RAxML v8.2.12, using the GTR+Gamma distribution model at the CIPRES science gateway (49) and found a high correlation between both the trees (NJ vs ML) (Mantel statistic $r = 0.935$, $P < 0.001$, method = spearman, 999 permutations). Finally, we conducted a similar analysis for each individual study (building an NJ phylogenetic tree using all OTUs present and running ConsenTRAIT) to confirm that the cross-study results were not starkly different when using all OTUs within a study.

## RESULTS

We screened over 1,500 abstracts of published literature and obtained data for 21 fiber diet interventions (from 12 studies) performed in healthy humans, for a total of 2,564 samples from 538 subjects (Tables 1 and 2). The duration of interventions ranged from 3 days to 84 days (Mdn = 15.5 days; SD = 21.3 days; Table 2) with a minimum of two fecal collection timepoints (before and after the diet intervention) but some collected up to eight times. While we included as many types of fiber interventions as possible, they were dominated by alpha-glycan and fructan supplements (Table 2).

## Alpha-diversity responses

We used three metrics to quantify changes in alpha-diversity: Shannon index, Simpson index, and observed richness. Short-term increases in dietary fiber consumption resulted in a highly consistent decline in bacterial alpha-diversity across studies. Richness tended to decrease in all but one intervention using starch-entrapped microspheres (Rasmussen_2017_V1V3_SM12) with ten interventions showing a statistically significant decline (paired-*t*-test $P < 0.05$; Table 3; Fig. 1). Five interventions showed a significant decline in bacterial alpha-diversity with at least two of the metrics used and eleven with at least one (paired-*t*-test $P < 0.05$; Table 3). Additionally, in all 21 interventions, at least two of the three alpha-diversity metrics decreased, even if not significantly (Fig. 1; Table 3).

## Beta-diversity responses

Increased fiber intake also had a consistent effect on gut microbiome beta-diversity in healthy humans. As expected, inter-individual variation in microbiome composition was high. Microbiome composition differed significantly among individuals in every study, and on average, explained 82% of the compositional variation observed (PERMANOVA: $P < 0.05$; Table 3). Despite this variability, in 14 out of 21 studies, a significant effect of the fiber intervention on microbiome composition was still detected. Notably, five of the seven non-significant interventions used less digestible fibers (cross-linked resistant starches, psyllium) and whole foods. Conversely, four other cross-linked resistant starch interventions led to a significant beta-diversity response. Of the significant beta-diversity responses, the interventions explained a relatively small but consistent amount of

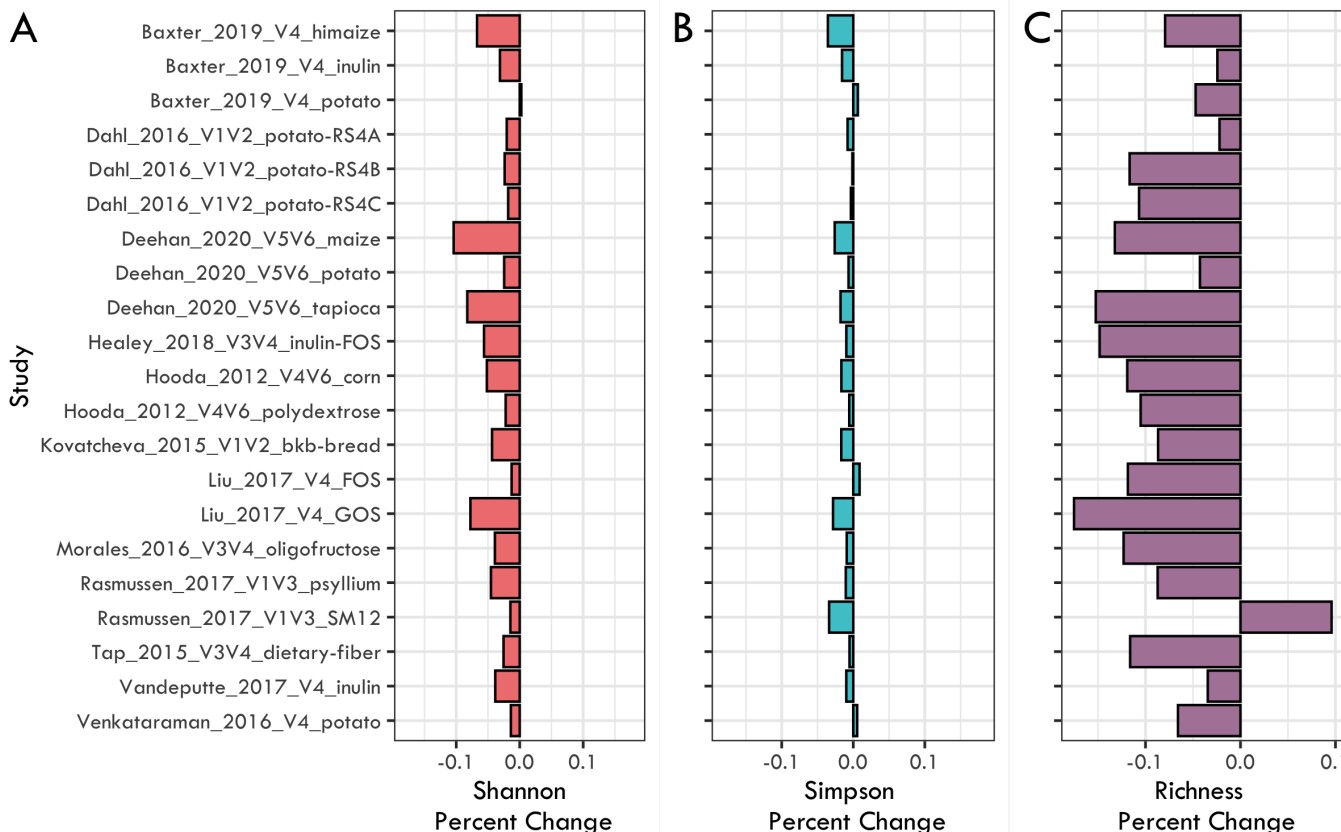

**FIG 1** Percent change for alpha diversity metrics: (A) Shannon index, (B) Simpson index, and (C) Richness. Alpha diversity metrics were calculated using ESVs and rarefied data (see Materials and Methods for details). Percent change was measured by subtracting the before-fiber intervention mean from the after-fiber intervention mean. When multiple timepoints where available, only the first and the last were used for paired-data points. See Table 3 for statistical significance of changes by study.

microbiome variation across studies, ranging from 0.2% to 4.6%, for an average of 1.5% of compositional variation (PERMANOVA: $P < 0.05$; Table 3). To further investigate the change in microbiome composition, we tested for differences in the compositional variation between the "before" and "after" fiber intervention groups in each study. Only one intervention (the Deehan_2020_V5V6_maize-RS4) showed a significant difference in compositional variability. Thus, the influence of the fiber interventions (the % variability explained) appears to be due to shifts in the average composition of the gut microbiome, rather than changes in compositional variability after the fiber intervention (Betadisper analysis, Table S1).

## Phylogenetic responses

To detect specific taxa (OTUs) and broader phylogenetic clades that consistently shifted after fiber interventions across studies, we re-analyzed only a subset of the interventions that amplified the same 16S rRNA region. This subset included fiber inventions involving the following: resistant corn starch, inulin, resistant potato starch, polydextrose, FOS, GOS, oligofructose, and high fiber meals (Table 4). After averaging the log$_2$-fold change responses for the widespread OTUs, we identified five bacterial OTUs that displayed significant, highly positive responses to fiber interventions (log$_2$-fold change $> 1$). The positive responding taxa belonged to the families *Bifidobacteriaceae* (three from *Bifidobacterium* genus, phylum Actinobacteriota), *Burkholderiaceae* (one from *Sutterella* genus, Proteobacteria), and *Ruminococcaceae* (one from *Faecalibacterium* genus, phylum Firmicutes). Among these taxa, OTUs belonging to the *Bifidobacteriaceae* family had the highest positive response to fiber with an average of 1.3 positive log$_2$-fold change, followed by *Burkholderiaceae* and *Ruminococcaceae* with 1.2 and 1.1 log$_2$-fold change, respectively. We also identified eight bacterial taxa that showed a highly negative response to fiber treatment (log$_2$-fold change $< -1.0$). These taxa all fell within the class Clostridia (phylum Firmicutes) and belonged to the following families: CAG-508 (three from UMGS1994, CAG-354, and unidentified genus), *Lachnospiraceae* (one from *Mediterraneibacter* and three from unidentified genus), and *Ruminococcaceae* (one from *Negativicutes* genus). The OTUs belonging to the *Lachnospiraceae* family had the strongest negative log$_2$-fold change with an average of $-1.4$, followed by CAG-508 and *Ruminococcaceae* with $-1.2$ and $-1.1$ log$_2$-fold change, respectively (Fig. 2).

We next identified broader phylogenetic clades whose response to the fiber intervention was conserved and calculated the average phylogenetic depth ($\tau_D$) of conservation. The three most predominant phyla in the phylogenetic tree were Firmicutes, Bacteroidota, and Actinobacteriota. Bacterial responses, positive and negative, to fiber treatment were significantly conserved with an average phylogenetic depth, $\tau_D$, of 0.019 and a 0.020 16S rRNA distance, respectively (permutation test; $P < 0.05$, Fig. 3). However, not all groups within a phylum responded in the same manner. For example, not all Actinobacteriota responded positively. Further, these patterns held within the individual interventions. The depth at which the fiber responses were conserved was greater than expected given a randomized distribution ($P < 0.05$) for all studies except Liu_2017_V4 (Table 4). On average, the degree of conservation for positively responding clades was of $\tau_D = 0.021$ ($n = 6$ significant interventions) and for negatively responding clades was $\tau_D = 0.019$ ($n = 4$ significant interventions) (Table 4). We note that these results are limited to inventions with fiber supplements mostly derived from alpha-glycans and fructans. Further studies are needed to assess whether these responses would be similar for other fiber types.

## DISCUSSION

Our re-analysis of bacterial 16S rRNA data from fiber intervention studies in healthy humans demonstrates that short-term increases in fiber consumption result in remarkably consistent responses in bacterial alpha-diversity, compositional variation (beta-diversity), and average changes in the relative abundance of some particular taxa despite a myriad of study differences, including the fiber type and amount and experimental

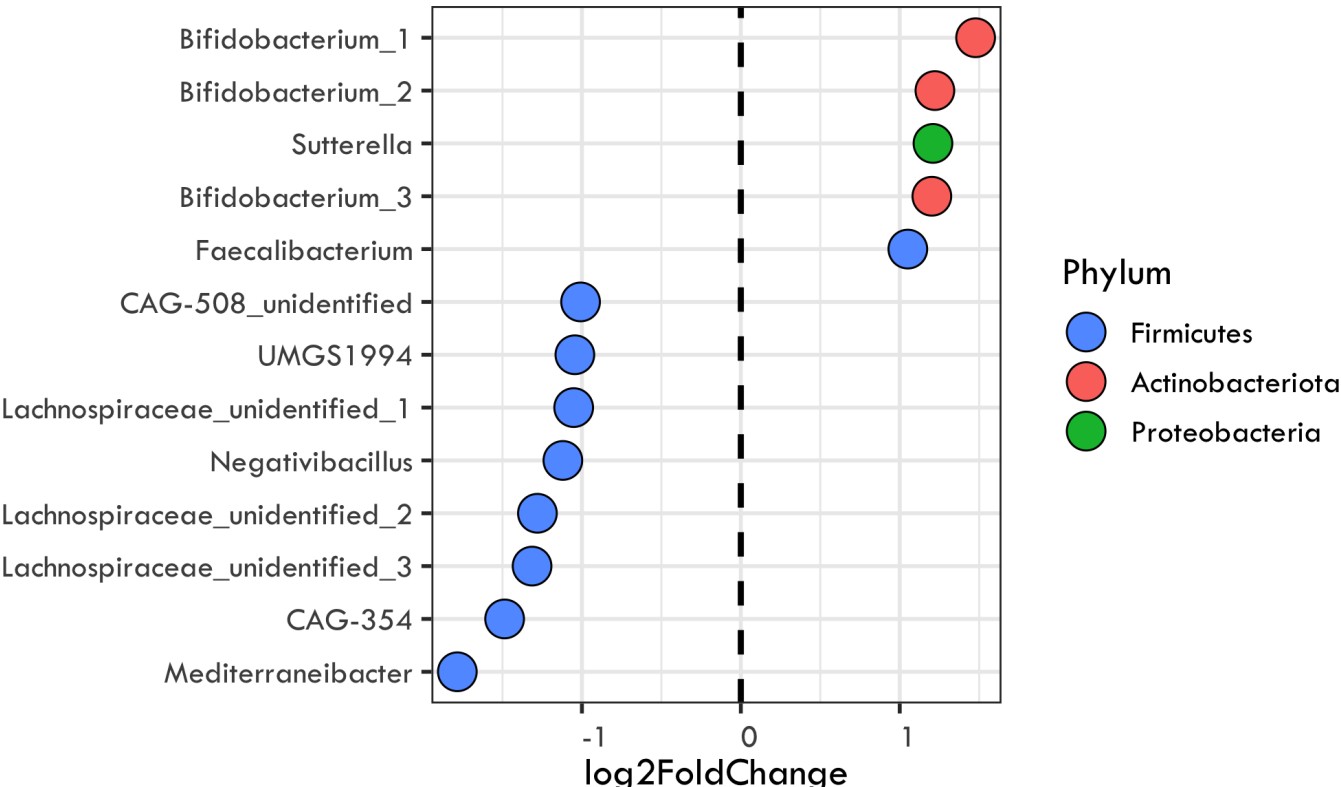

**FIG 2** Top bacterial responders to fiber interventions. Each point represents a bacterial clade that had a large response (>1 or <−1) in abundance based on the averaged log$_2$-fold changes calculated by DESEq2 of widespread OTUs (present in at least three studies). The data points are identified by their genus or, when unknown, as family + "_unidentified" following GTDB classification.

duration. Furthermore, bacterial responses were phylogenetically conserved, allowing us to identify bacterial clades that generally increased or decreased across studies. Thus, even though individuals may vary in the specific taxa (OTUs) that they carry, taxa within these clades tended to respond similarly across studies to the fiber types included in this analysis. Our results may seem somewhat surprising given well-known fiber-specific responses of the gut microbiome (50, 51) and high inter-individual microbiome variability. However, these observations are not necessarily in conflict. First, our results capture an average response, across individuals, studies, and broadly defined OTUs (97% 16S rRNA sequence similarity) and do not preclude variation within these categories. Second, we focus on the largest responders (those with the highest fold changes). These particular taxa are likely the most consistent responders, and more variable taxa would be less likely to emerge from the phylogenetic analyses. Finally, as mentioned in the Introduction, our analysis is limited by the range of fiber types used in the studies. Additional studies are needed to assess whether the taxa identified respond consistently to additional fiber types.

In line with previous work (11, 14, 24, 50–54), increased fiber tended to reduce alpha-diversity across all interventions. We therefore conclude that a sudden increase in fiber intake, regardless of the fiber type, generally decreases bacterial alpha-diversity. Previously, it has been suggested (52, 55) that such a decline in alpha-diversity after a fiber intervention could be due to the short-term nature of the interventions. Specifically, short-term studies might capture only a transitional period, where bacteria that are not well adapted to the changing environment (e.g., decreased pH due to increased fermentation) decline in relative abundance relative to taxa that can quickly consume the newly available carbohydrates. This reasoning suggests that over a longer time period, bacterial alpha-diversity might decline less (or perhaps even increase) as more slowly

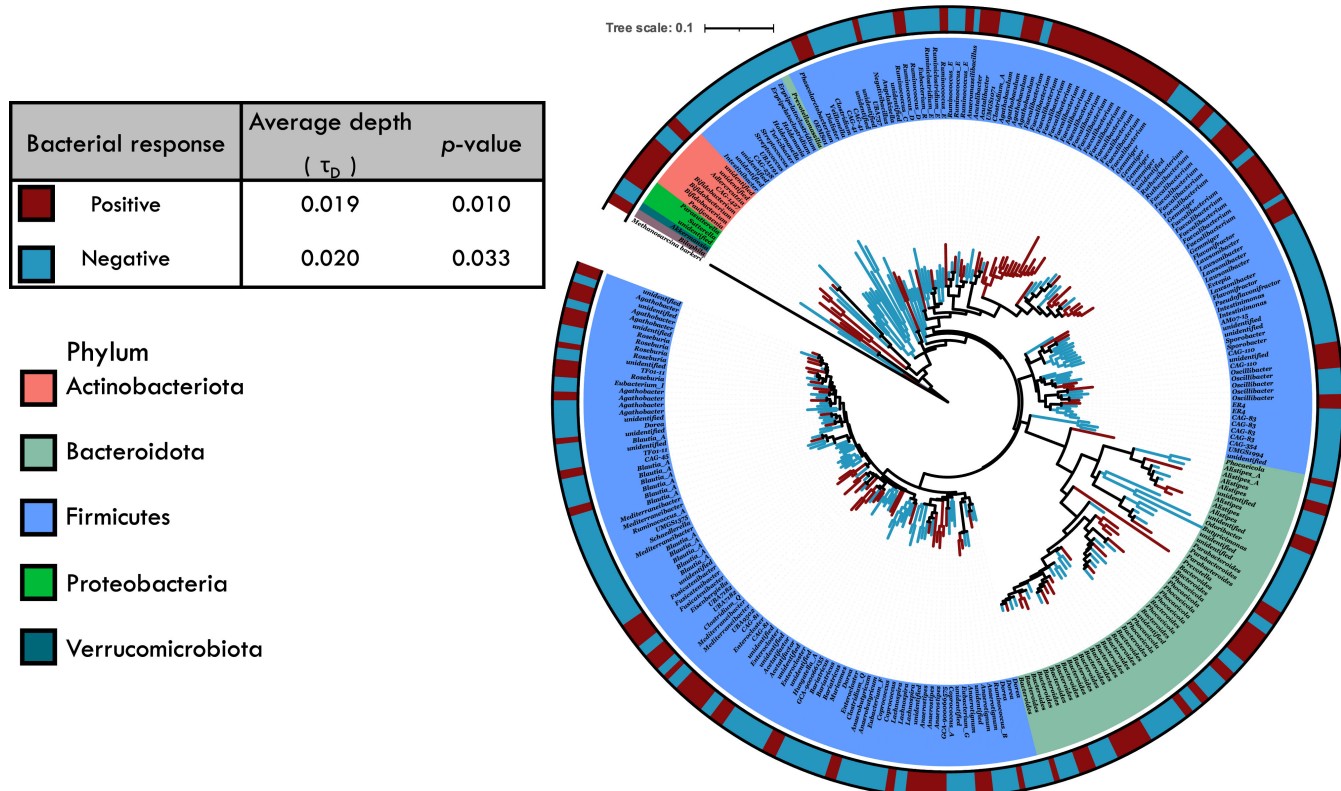

**FIG 3** Phylogenetic distribution of the averaged responses to fiber intervention. The widespread OTUs (present in at least three studies) are colored based on their response to fiber, outermost ring and branch with red = positive or blue = negative. The inner ring represents the phylum-level taxonomy of the OTUs determined using the GTDB trained database. The average depth of conservation and *P*-values are shown in the inset panel.

growing fiber consumers increase in relative abundance. In the studies analyzed here, the alpha-diversity response was not correlated with intervention length (Shannon's Spearman $r = -0.156$, $P > 0.05$; Simpson's Spearman $r = -0.325$, $P > 0.05$), but we note that the studies ranged from only 3 to 84 days. Thus, longer studies are needed to investigate the effects of increased dietary fiber on the long-term dynamics of gut microbiome alpha-diversity.

The changes in the overall variation in bacteria composition (beta-diversity) were also similar among studies. While fiber intervention explained a relatively small amount of compositional variation compared to interindividual variability (1.5% vs 82%), a significant effect of the fiber intervention on microbiome composition was detected in 14 out of 21 studies. Notably, the number of subjects in the non-significant studies included <34 individuals (although some studies with less than that amount did find significant effects), suggesting that the studies were statistically underpowered. In contrast, fiber interventions with 40–50 individuals detected even small (0.2%–0.7 %) effects on microbiome composition.

The effects on overall bacterial beta-diversity were largely driven by changes in the relative abundance of well-known fiber degrading taxa. OTUs belonging to the genus *Bifidobacterium* showed the strongest positive response across fiber interventions (Fig. 2). Further, these responses were phylogenetically conserved, meaning that the response was not limited to particular strains of *Bifidobacterium,* but seems to be a response that is shared across the genus. Indeed, the genus has been previously found to increase in abundance following an increased fiber intake (11, 12, 24). Bifidobacteria possess a high number of carbohydrate active enzymes (CAZymes) that allow the degradation of various plant carbohydrates (56–58) and thus its ability to respond to increased fiber availability is not surprising. These fiber degrading bacteria are thought to benefit health

via cross-feeding interactions with butyrate-producing colon bacteria and other short chain fatty acid (SCFA) producers (59, 60). Indeed, fiber rich diets are also associated with positive changes in SCFAs (61, 62).

Our analysis also detected responses to fiber intake in the *Sutterella* and *Faecalibacterium* genera. OTUs within the *Sutterella* genus increased significantly across the studies, although we could only identify one previous report of this response in a fiber intervention of pregnant women suffering from hypertensive disorders (63). *Sutterella* species have been associated with health disorders such as autism and metabolic syndrome (64, 65), but are also present in healthy humans. Their ability to adhere to intestinal epithelial cells might indicate a positive relationship with its host (66), but further investigations into the role of this genus, particularly in terms of fiber degradation, is warranted. Similar to our study, some species within this genus (e.g., *F. prausnitzii*) have been found to increase in response to fiber intake (67–69), and a lower relative abundance of *Faecalibacterium* has been associated with diseases such as inflammatory bowel disease (IBD), colorectal cancer, and dermatitis (70–72).

We also identified specific taxa and clades that consistently decreased during the fiber interventions. While a significant positive response would seem to indicate the use of fiber as a carbon resource, it is less clear what a consistent negative response means. As mentioned above, one possibility is that increased fiber degradation will change the gut environment (e.g., low pH due to increased fermentation) and some taxa might not compete as well in these conditions, hence decreasing their abundance. All negative responding taxa fell within the class Clostridia (phylum Firmicutes) with the *Lachnospiraceae* family showing the strongest negative response. While this family is typically found in the human gut microbiome and some members are main producers of SCFAs (73), studies have suggested that members of this family may be associated with certain diseases (74). However, out of the four responding taxa within this family, we were only able to identify the *Mediterraneibacter* genus. Previous research has found that *Mediterraneibacter* is associated with host obesity in women with polycystic ovary syndrome (75) and that is able to produce aldehyde alcohols which are considered harmful to the host (73), whereas its role in fiber fermentation has not been described. Together, the identification of both positively and negatively responding clades provide candidates for investigating the mechanistic links between a fiber-rich diet, the metabolic outputs of fiber degradation, and intestinal health.

Finally, although microbial responses to fiber interventions are thought to be highly individualized to the person (58, 76), bacterial taxa that respond to fiber interventions showed a phylogenetic signal. Specifically, bacterial taxa that respond positively or negatively to fiber intake exhibited a significant average phylogenetic depth of conservation ($\tau_D = 0.021$ and $\tau_D = 0.019$ ; $P < 0.05$; Fig. 3). Depth of conservation ($\tau_D$) serves as a metric for predicting the distribution of functional traits in microorganisms (77). The depth of the fiber response was similar to that previously found for nitrogen fixation traits ($\tau_D = 0.018$–$0.020$), but more deeply conserved than that of simple carbon utilization ($\tau_D = 0.011$) (27, 77, 78) and the ability to produce extracellular enzymes ($\tau_D = 0.008$–$0.01$) (79). Moreover, the average depth of bacterial responses to fiber intervention displayed a relatively narrow range across studies ($\tau_D = 0.014$–$0.028$; Table 4).

These results come with certain limitations inherent to the use of 16S rRNA data and the re-analysis of publicly available data. First, phylogenetic trees built with 16S rRNA amplicon sequences are not as reliable as multi-locus trees (80); however, they are still useful to estimate the depth of the response to fiber interventions and to compare this response with other traits that have been analyzed previously (26, 27, 77). Second, combining data from distinct studies resulted in unequal sample sizes across fiber interventions, hindering comparisons between fiber types. In the future, studies that directly compare different fibers would be useful to test for variation in bacterial responses to particular fiber types. This would be expected as some gut bacteria are known to specialize on different types of fibers (81–83).

## Conclusion

We showed that a phylogenetic approach, that has been previously used to test bacterial trait conservation in environmental samples (26, 27, 77), can be useful to disentangle the bacterial responses to a dietary change in the human gut microbiome. Despite the high microbial variation in human subjects, this method can be applied to human related microbiomes to identify bacterial clades that are generally responsive to dietary changes and their average phylogenetic depth of conservation. Similar types of microbiome data syntheses could be useful for investigating compositional responses of the gut microbiome to other types of interventions or diseases. Our results support a previous analysis of 28 studies (including 11 different diseases) that showed that not only does disease-state generally alter the gut microbiome, but that independent studies generally see consistent responses, and that these responses differ by type of disease (84)

Finally, we observed that the individual variation of gut microbiome composition is high, on average 82% as found here; therefore, it is important to put any intervention effect into perspective. Indeed, it would seem very unlikely, once methodological error is accounted, to find a treatment effect that explains more than single digits (22). This highlights that even relatively small effect sizes are not necessarily unimportant. Within a person, compositional shifts in the gut such as those caused by increased dietary fiber may be consequential for gut functioning relative to background fluctuations. These results also highlight the benefit of using cross-over study designs in diet interventions to help to detect the relatively subtle effects of such interventions on an individual's microbiome (85).

## ACKNOWLEDGMENTS

We thank all the authors of the studies mentioned here for making their data available for re-analysis. We also thank Dr. Katrine Whiteson and Kristin Barbour for their feedback on the manuscript and the members of the J. Martiny lab for their encouragement.

This work was supported by NIH T32AI141346, the UCI Faculty Mentor Program, a UC President's Dissertation Year Fellowship, and the Rose Hills Foundation Science & Engineering Fellowship.

J.B.H.M. and C.I.R. conceived the project, wrote the manuscript, and interpreted the data. C.I.R. collected data and conducted the bioinformatic analyses. K.I. helped with bioinformatic analysis and data interpretation. All authors read and approved the final manuscript.

## AUTHOR AFFILIATIONS

[1]Department of Ecology and Evolutionary Biology, University of California, Irvine, California, USA
[2]Institute of Ecology, College of Urban and Environmental Sciences, Peking University, Beijing, China

## AUTHOR ORCIDs

Cynthia I. Rodriguez http://orcid.org/0000-0003-2772-9013
Jennifer B. H. Martiny http://orcid.org/0000-0002-2415-1247

## FUNDING

| Funder | Grant(s) | Author(s) |
| --- | --- | --- |
| HHS | National Institutes of Health (NIH) | NIH T32AI141346 | Cynthia I. Rodriguez |
| UCI Faculty Mentor Program | | Cynthia I. Rodriguez |
| Rose Hills Foundation (RHF) | | Cynthia I. Rodriguez |

## AUTHOR CONTRIBUTIONS

Cynthia I. Rodriguez, Conceptualization, Data curation, Formal analysis, Investigation, Visualization, Writing – original draft, Writing – review and editing | Kazuo Isobe, Data curation, Formal analysis, Writing – original draft | Jennifer B. H. Martiny, Conceptualization, Investigation, Supervision, Writing – original draft, Writing – review and editing

## DATA AVAILABILITY

All data generated or analyzed during this study are included or referenced in this published article and its additional information files. The accession numbers for the raw 16S rRNA sequencing reads can be found in Table 1. The processed reads, OTU tables, and metadata files supporting the conclusions of this article can be found in Figshare repository https://doi.org/10.6084/m9.figshare.21295352. All bioinformatic pipelines and code can be found at the following Github repository: https://github.com/cirodri1/fiber-data_records.

## ADDITIONAL FILES

The following material is available online.

### Supplemental Material

**Table S1 (mSystems00133-24-s0001.docx).** Betadisper results.

### Open Peer Review

**PEER REVIEW HISTORY (review-history.pdf).** An accounting of the reviewer comments and feedback.

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
