## [Reviewer comments · mSystems]

Short-term dietary fiber interventions produce consistent gut microbiome responses across studies

Cynthia Rodriguez, Kazuo ISOBE, and Jennifer Martiny

Corresponding Author(s): Cynthia Rodriguez, University of California Irvine

Review Timeline:

Submission Date:	January 29, 2024
Editorial Decision:	March 5, 2024
Revision Received:	March 10, 2024
Accepted:	April 15, 2024

Editor: Emily Cope

Reviewer(s): The reviewers have opted to remain anonymous.

Transaction Report:

DOI: <https://doi.org/10.1128/msystems.00133-24>

Re: mSystems00133-24 (Short-term dietary fiber interventions produce consistent gut microbiome responses across studies)

Dear Dr. Cynthia Itzel Rodriguez:

Thank you for the privilege of reviewing your work. Below you will find instructions from the mSystems editorial office, and the reviewer comments. While the manuscript has been improved, there remain some critiques to be addressed.

Revision Guidelines

Sincerely,
Emily Cope
Editor
mSystems

Reviewer #3 (Comments for the Author):

Overall, the analysis was good, although there are a few things I noticed that might need to be corrected as the data from other papers in Table 3 is wrong in a few places. Also, the authors casually use terms which are not accurate in this context (OTU table to mean ESV table, or microbiome to mean bacterial community). It's common in this research field to misuse those terms, but I encourage the authors to be more precise here because 16S analysis is heavily dependent on the way the data is generated and processed. The amount of information that was gained from a meta-analysis of this size is pretty minimal. I would

have expected twice and many figures and much more interpretation on trends.

The authors appear to have put significant work into addressing the previous review.

lines 26, 33, and throughout: correct to "gut bacterial communities" when referring to the data used here. Only 16S was used and this does not provide enough data to describe the microbiome.

line 27: reduced alpha diversity? Fiber increases alpha diversity in about 75% of people. I think you are basing this conclusion on Shannon Index which is not accurately reporting changes in the community. You should use richness and evenness, instead of Shannon, because they often display opposite patterns that make Shannon values lower and obscure trends.

line 148 and 157 and 165 and throughout the Methods: I would rephrase "OTU table" to "sequence table" to avoid confusion, even though the DADA2 and other programs still use that term. ASV/ESVs are created very differently from OTUs, which the authors clearly state in the Intro, but since the Methods section of this paper is likely to be very appealing to readers, some may skip straight to this section and miss the nuance.

Line 291, Results section, and Table 4: OTUs are not specific taxa. Do you mean to say ESVs combined into genus categories? Throughout the results, you should not be using OTU unless you mean a grouping defined by mathematical cutoffs rather than taxonomic grouping, which is what you seem to be referring to. Even your ConsenTRAIT analysis is looking at shared grouping but is not referring to genetic grouping, so Table 4 should say ESVs.

Figure 1: I think showing richness and evenness would be much more useful than Shannon diversity which obscures trends by combining those two metrics. If richness and evenness have opposing patterns, it will sometimes cancel out or make Shannon look lower than the amount of richness would suggest.

Table 3: The Baxter 2019 paper reports no change in alpha diversity, and they used Simpson. In your Table 3, you report a significant decrease, as well as a non-significant one using Shannon. I think you might have mixed up which data is from which paper, and you need to add a "no change" category under alpha diversity.

Reviewer #4 (Comments for the Author):

The authors in my opinion have responded in a satisfactory manner to the reviewers' critiques.

Response to Reviewers

Manuscript Number: mSystems00133-24

Manuscript Title: Short-term dietary fiber interventions produce consistent gut microbiome responses across studies

Dear Editor,

We thank the editor and the reviewers for their time and for providing comments to improve our manuscript “Short-term dietary fiber interventions produce consistent gut microbiome responses across studies” (mSystems00133-24). We have carefully reviewed each comment and addressed them in a point-by-point manner below. Responses are shown in blue with specific changes to the manuscript noted in red.

We hope that you find this revised manuscript suitable for publication at *mSystems*. We look forward to hearing from you.

Sincerely,

Cynthia Rodriguez, on behalf of all authors

Reviewer #3:

Comment 1: Overall, the analysis was good, although there are a few things I noticed that might need to be corrected as the data from other papers in Table 3 is wrong in a few places. Also, the authors casually use terms which are not accurate in this context (OTU table to mean ESV table, or microbiome to mean bacterial community). It's common in this research field to misuse those terms, but I encourage the authors to be more precise here because 16S analysis is heavily dependent on the way the data is generated and processed. The amount of information that was gained from a meta-analysis of this size is pretty minimal. I would have expected twice as many figures and much more interpretation on trends.

The authors appear to have put significant work into addressing the previous review.

> We thank the reviewer for their comments and for noticing our efforts to address the previous reviews. We address the Reviewer’s specific comments in detail below.

Comment 2: lines 26, 33, and throughout: correct to "gut bacterial communities" when referring to the data used here. Only 16S was used and this does not provide enough data to describe the microbiome.

> We agree with the reviewer that the term “microbiome” needs to be better defined in our manuscript as we are not assessing viruses, fungi, and archaea. When appropriate, we have changed the word microbiome for “gut bacterial communities,” particularly in the Abstract (Lines 26, 33, 45-45, 156). Additionally, we define the term “microbiome” the first time we use it as it refers to our study (Lines 46-48).

Comment 3: line 27: reduced alpha diversity? Fiber increases alpha diversity in about 75% of people. I think you are basing this conclusion on Shannon Index which is not accurately reporting changes in the

community. You should use richness and evenness, instead of Shannon, because they often display opposite patterns that make Shannon values lower and obscure trends.

>The reviewer's statement that "fiber increases alpha diversity in about 75% of people" is not our impression of the literature. It is because of these opposite impressions from different researchers that we decided there was a need for a systematic analysis of the literature.

>Beyond our results, reports of a decrease in alpha-diversity after a dietary-fiber intervention is not uncommon. We have now added citations (8) of other studies describing decreased alpha-diversity. We also provide possible reasons for such a decrease after a fiber intervention (Lines 355-368).

> We agree that one metric of alpha-diversity is not enough to assess important patterns in a microbial population. For that reason, we already included two metrics, Shannon and Simpson (Table 3). Per the reviewer's suggestion, we have now also added Richness (Lines 173-178). The Richness metric supports our statement that alpha-diversity, on average, decreases across fiber interventions (Table 3, Figure 1). To make it easier for the reader to see differences across these alpha-diversity metrics, we have included three columns in our Table 3 under the Alpha-diversity to report all three metrics and expanded Figure 1 to include Richness.

Comment 4: line 148 and 157 and 165 and throughout the Methods: I would rephrase "OTU table" to "sequence table" to avoid confusion, even though the DADA2 and other programs still use that term. ASV/ESVs are created very differently from OTUs, which the authors clearly state in the Intro, but since the Methods section of this paper is likely to be very appealing to readers, some may skip straight to this section and miss the nuance.

> We have rephrased "OTU table(s)" when referring to our tables coming from the DADA2 platform to "ESV table(s)" (Lines 149-150, 156, 158, 166, 189).

> When referring to the data used for the ConsenTRAIT analyses, the sequences were clustered at 97% identity; hence, those sequences are considered OTUs by the reviewer's definition. We explain this in the Methods section (Lines 217-218).

Comment 5: Line 291, Results section, and Table 4: OTUs are not specific taxa. Do you mean to say ESVs combined into genus categories? Throughout the results, you should not be using OTU unless you mean a grouping defined by mathematical cutoffs rather than taxonomic grouping, which is what you seem to be referring to. Even your ConsenTRAIT analysis is looking at shared grouping but is not referring to genetic grouping, so Table 4 should say ESVs.

> We apologize for the confusion caused by OTU vs ESVs. As mentioned above, we now changed the wording for the tables obtained from DADA2 to ESV tables" (Lines 149-150, 156, 158, 166, 189), while keeping OTU for the taxonomic units used in the ConsenTRAIT analysis (i.e., those clustered by 97% sequence identity) (Lines 217-223).

> Table 4 refers to the 97% genetic grouping of sequences; hence, we keep OTU here. We clarify this in our Methods and in Table 3's description.

Comment 6: Figure 1: I think showing richness and evenness would be much more useful than Shannon diversity which obscures trends by combining those two metrics. If richness and evenness have opposing patterns, it will sometimes cancel out or make Shannon look lower than the amount of richness would suggest.

> We thank the reviewer for this recommendation, which further strengthens our conclusions. As mentioned in our previous comment, we now include Shannon (an index that takes into account both abundance and evenness), Simpson (a dominance index that gives more weight to abundant taxa), and Richness (a count of taxa present) in our analyses (Table 3, Figure 1, Lines 271-279).

> It is worth highlighting that Richness showed a decreasing pattern after every fiber intervention except one (Rasmussen_2017_V1V3_SM12).

Comment 7: Table 3: The Baxter 2019 paper reports no change in alpha diversity, and they used Simpson. In your Table 3, you report a significant decrease, as well as a non-significant one using Shannon. I think you might have mixed up which data is from which paper, and you need to add a "no change" category under alpha diversity.

> Thank you for noticing the difference in results between the original publication and our findings. We have checked multiple times that the datasets and our results are correct. The original Baxter 2019 study reported no "significant" change in alpha-diversity by *inverse Simpson* index. They do not seem to report the direction of this trend. It is important to note that it would be possible for us to obtain different results from the original study without making an error. We do not include all the treatments and timepoints from each study, only the ones needed for consistent comparisons across the studies (two time points for a paired t-test and percent change analysis, separately for each fiber type). Also, we use consistent bioinformatic analyses that might change the results. However, in the case of Baxter, our results are not very different from the ones in the original paper. We only found that hi-maize induced a statistically significant decrease, while the other two interventions (inulin and potato) show no statistical difference. We now highlight that discrepancies between the original studies and ours are possible in the Methods (Lines 202-205).

> To be as clear and transparent as possible we offer a link to a Github page where we include all the bioinformatic pipelines used and our code for replication (Lines 709-710). We also published a Data Description paper where we provide the curated datasets and explain how they were processed in detail (Reference # 28; Lines 139-141).

Reviewer #4:

Comment 1: The authors in my opinion have responded in a satisfactory manner to the reviewers' critiques.

> We thank the reviewer for taking the time reading our manuscript and responses.

Re: mSystems00133-24R1 (Short-term dietary fiber interventions produce consistent gut microbiome responses across studies)

Dear Dr. Cynthia Itzel Rodriguez:

Your manuscript has been accepted, and I am forwarding it to the ASM production staff for publication. The second reviewer has one minor comment to be addressed, which will clarify where your results diverge from the original report. Your paper will first be checked to make sure all elements meet the technical requirements. ASM staff will contact you if anything needs to be revised before copyediting and production can begin. Otherwise, you will be notified when your proofs are ready to be viewed.

Cover Image Submissions: If you would like to submit a potential Cover Image, please email a file and a short legend to msystems@asmusa.org. Please note that we can only consider images that (i) the authors created or own and (ii) have not been previously published. By submitting, you agree that the image can be used under the same terms as the published article. Image File requirements: TIF/EPS, 7.5 inches wide by 8.25 inches tall (at least 2,250 pixels wide by 2,475 pixels tall), minimum 300 dpi resolution (600 dpi preferred), RGB, and no figure elements, e.g., arrows or panel labels. The legend should be a short description of the image, 1-2 sentences recommended.

We recognize that the video files can become quite large, so to avoid quality loss ASM suggests sending the video file via <https://www.wetransfer.com/>. When you have a final version of the video and the still ready to share, please send it to mSystems staff at msystems@asmusa.org.

Sincerely,

Emily Cope
Editor
mSystems

Reviewer #6 (Comments for the Author):

I once again admire the care the authors have taken to respond to prior critiques.

In reviewing the last comment in the Response to Reviewer #3 (as well as looking back over Baxter 2019 myself), I can see where Reviewer #3's concerns arise from. It would be unfortunate (and I suspect not the authors' intention) for readers to misconstrue the findings and interpretations in the original publications from which these data were drawn.

To alleviate such concern, I might suggest that the authors modify the alpha diversity columns in Table 3 to include a small indicator (e.g. carat or pound sign?) denoting when the authors' analyses appear to diverge from the original papers'.